# Masked Unsupervised Self-training for Label-free Image Classification

**Junnan Li, Silvio Savarese, Steven Hoi**
Salesforce Research
{junnan.li,ssavarese,shoi}@salesforce.com

## Abstract

State-of-the-art computer vision models are mostly trained with supervised learning using human-labeled images, which limits their scalability due to the expensive annotation cost. While self-supervised representation learning has achieved impressive progress, it still requires a second stage of finetuning on labeled data. On the other hand, models pre-trained with large-scale text-image supervision (e.g., CLIP) have enabled zero-shot transfer to downstream image classification tasks. However, the zero-shot performance of CLIP-like models are often insufficient for real-world adoption. In this paper, we aim to leverage the abundant unlabeled data from a target domain to improve the performance of a pre-trained zero-shot classifier, by unsupervised finetuning of the pre-trained model. We propose Masked Unsupervised Self-Training (MUST), a new unsupervised adaptation method which leverages two different and complementary sources of training signals: pseudo-labels and raw images. MUST jointly optimizes three objectives to learn both class-level global feature and pixel-level local feature and enforces a regularization between the two. We demonstrate the efficacy of MUST on a variety of downstream tasks, where it improves upon CLIP by a large margin. MUST also outperforms supervised few-shot adaptation methods. It achieves a top-1 accuracy of 77.7% on ImageNet using ViT-B, +9.4% higher than CLIP, and +6.2% higher than 16-shot CLIP adaptation. Our code is available at `https://github.com/salesforce/MUST`.

## 1 Introduction

Zero-shot image classification is a challenging goal that marks the capability of a vision model to solve tasks without human supervision. Recently, vision-language pre-training (e.g. CLIP (Radford et al., 2021)) has shown promising performance on open-vocabulary zero-shot classification, where it leverages web-scale image-text pairs to train image and text encoders that can be transferred to downstream tasks through natural language prompting. However, the zero-shot performance of CLIP is often inadequate for real-world adoptions, especially when compared to models that are trained with supervised learning. On the other hand, there are abundant unlabeled data available for many tasks. In this paper, we aim to improve the performance of an open-vocabulary zero-shot classifier by finetuning it on unlabeled images from a downstream task.

Given unlabeled data, the key question is: *what is the source of supervision?* Numerous papers have attempted to answer this question. Among them, *self-training* and *self-supervised learning* are two of the most dominant approaches. In self-training (Zoph et al., 2020; Xie et al., 2020; Lee et al., 2013), pseudo-labels are generated by a teacher model and then used to supervise task-specific training of a student model, where the student usually has an equal model size as the teacher. On the other hand, self-supervised learning methods are generally task-agnostic. Masked image modeling (Bao et al., 2022; He et al., 2021; Xie et al., 2022), which trains the model to predict missing information from masked image patches, has recently emerged as the superior self-supervised learning method for vision transformers (ViT) (Dosovitskiy et al., 2021). However, both self-training and self-supervised learning have their limitations. Self-training overly relies on the pseudo-labels as the only source of supervision, thus is prone to overfitting to the noise in pseudo-labels. Self-supervised learning requires an additional stage of task-specific finetuning on labeled data, thus is not a one-stop solution.

In this paper, we propose Masked Unsupervised Self-Training (MUST), a simple and effective method for label-free image classification. MUST performs unsupervised learning using both pseudo-labels and raw images as two different and complementary training signals. Specifically, MUST jointly

optimizes three objectives to finetune a pre-trained classification model (*e.g.* CLIP) on unlabeled images: (1) Self-training objective to learn global task-specific class prediction; (2) Masked image modeling objective to learn local pixel-level information; (3) Global-local feature alignment objective to bridge the knowledge learned from the two sources of supervision.

We validate the efficacy of MUST on 8 image classification tasks across a variety of domains, showing significant improvement over CLIP (Radford et al., 2021). MUST also outperforms supervised few-shot adaptation methods (Zhou et al., 2021; 2022). For instance, MUST achieves 77.7% top-1 accuracy on ImageNet, +9.4% higher than CLIP, and +6.2% higher than 16-shot CLIP adaptation. On certain domains, MUST can achieve comparable performance with a fully-supervised method. We further perform extensive quantitative and qualitative analysis to examine the effect of each proposed component. MUST is a low-cost solution for image classification that unlocks the potential of CLIP-like models for practical scenarios where images are abundant but labels are scarce.

## 2  RELATED WORK

**Zero-shot learning** traditionally aims to recognize unseen classes by training the model on base classes (Xian et al., 2017; Wang et al., 2019), where the most common approach is to utilize auxiliary information such as attributes (Huynh & Elhamifar, 2020) or knowledge graphs (Wang et al., 2018). CLIP (Radford et al., 2021) popularizes a new approach for open-vocabulary zero-shot image classification by leveraging natural language supervision from web-scale datasets. Despite its impressive performance, the zero-shot accuracy of CLIP stills falls far below the supervised method in many domains. Some recent work tries to adapt CLIP to downstream tasks using labeled data (Zhou et al., 2021; Gao et al., 2021), which is less scalable compared to our unsupervised adaptation method. Our method is also orthogonal to CLIP-like research in model pre-training and can be applied to other image classification models (Jia et al., 2021; Li et al., 2022; Yao et al., 2022; Li et al., 2021), such as LiT (Zhai et al., 2022) which performs two-stage pre-training.

**Self-training** has shown promising progress in many domains including vision (Zoph et al., 2020; Xie et al., 2020; Sahito et al., 2022), NLP (He et al., 2020a), and speech (Kahn et al., 2020). Our method is more closely related to the self-training approaches proposed for semi-supervised learning (Tarvainen & Valpola, 2017; Sohn et al., 2020; Berthelot et al., 2020), where pseudo-labels on unlabeled data are used as training targets. We construct our self-training objective by following three principles: (1) *consistency regularization* (Sohn et al., 2020; Laine & Aila, 2017) which enforces the model to output the same prediction when the input is perturbed; (2) *entropy minimization* (Grandvalet & Bengio, 2004) which encourages the model to give "sharp" predictions with low entropy; (3) *prediction fairness* (Berthelot et al., 2020) which alleviates the model's bias towards certain classes. Our method leverages a separate supervision signal from raw image pixels to reduce the confirmation bias in self-training, which is orthogonal to methods for pseudo-label debiasing (Wei et al., 2021; Wang et al., 2022).

**Masked image modeling**, fueled by the success of vision transformers (Dosovitskiy et al., 2021), recently emerged as a more appealing self-supervised representation learning method over contrastive learning (He et al., 2020b; Chen et al., 2020; Hjelm et al., 2019). While some methods train the model to predict discrete tokens (Bao et al., 2022) or contextualized representations (Baevski et al., 2022) for masked image patches, MAE (He et al., 2021) and SimMIM (Xie et al., 2022) achieve competitive performance by simply predicting the pixel values. The MIM objective has also been used for test-time training (Gandelsman et al., 2022). Different from existing self-supervised learning methods which require an additional stage of supervised finetuning on labeled data, we synergically incorporate masked image modeling into self-training as a one-stage solution for zero-shot image classification.

## 3  METHOD

MUST is a simple unsupervised learning approach that adapts a pre-trained open-vocabulary classifier to a downstream task using unlabeled images. In this paper, we consider vision transformers pre-trained by CLIP (Radford et al., 2021) as the models to be adapted due to their distinctive zero-shot performance. CLIP pre-trains an image encoder and a text encoder with a contrastive loss such that paired images and texts have higher similarities in a shared embedding space. In order to perform zero-shot classification, CLIP converts a set of class names into text embeddings using ensemble of

Figure 1: On each downstream task, we convert CLIP's text embeddings into a linear classifier, which is fine-tuned together with the image encoder for unsupervised adaptation.

natural language prompts (*e.g.*, a photo of a {object}). During inference, it takes the dot-product between an image embedding and all text embeddings to produce the prediction logits for that image. As shown in Figure 1, we convert CLIP's non-parametric text embeddings into weights of a linear classifier, and directly finetune the linear classifier together with the image encoder for unsupervised adaptation.

Figure 2 shows the overall framework of MUST. Given an image, we follow ViT (Dosovitskiy et al., 2021) to divide it into regular non-overlapping patches. A `[CLS]` token is appended to extract global information, which is used by the classifier for prediction. We then randomly mask image patches by replacing a patch's embedding with a learnable `[MSK]` token. The output embeddings of `[CLS]` and `[MSK]` are used to jointly optimize three objectives: (1) global self-training, (2) local masked image modeling, and (3) global-local feature alignment. Next we delineate the details of each objective.

## 3.1 SELF-TRAINING WITH AN EMA TEACHER

The self-training objective is applied to the classifier's output. Given a batch of $B$ unlabeled images, we compute a pseudo-label for each image by passing a weakly-augmented version of the image to a teacher model. The teacher is parameterized by an exponentially moving average (EMA) (Tarvainen & Valpola, 2017; Grill et al., 2020) of the model parameters $\theta$. Specifically, the parameters of the EMA teacher $\Delta$ are intialized as $\theta$. During each update of $\theta$, $\Delta$ are updated with

$$\Delta = \mu\Delta + (1 - \mu)\theta \tag{1}$$

Let $q_b$ denote the EMA teacher's softmax prediction for the weakly-augmented image, we enforce a cross-entropy loss against the model's prediction $p_b$ for a strongly-augmented version of the same image:

$$\mathcal{L}_{\text{cls}} = \frac{1}{B} \sum_{b=1}^{B} \mathbb{1}(\max q_b \geq \tau) \text{H}(\hat{q}_b, p_b) \tag{2}$$

Following FixMatch (Sohn et al., 2020), we only use pseudo-labels with maximum scores above a threshold $\tau$, and convert the soft labels $q_b$ into "one-hot" hard labels by $\hat{q}_b = \arg\max(q_b)$.

Since the pseudo-labels generated by a CLIP model are often biased towards certain classes (Wang et al., 2022), minimizing $\mathcal{L}_{\text{cls}}$ alone would magnify the bias. To mitigate confirmation bias, we introduce a "fairness" regularization which encourages that on average, across a batch of samples, the model's prediction probability is close to a uniform distribution:

$$\mathcal{L}_{\text{reg}} = -\frac{1}{K} \sum_{k=1}^{K} \log(\bar{p}_k), \tag{3}$$

where $K$ is the total number of classes, and $\bar{p}$ is the model's average prediction across the batch. In cases where $K > B$, we compute $\bar{p}$ using moving average instead of batch average. We find this regularization to be beneficial even for datasets with long-tailed class distribution (see Table 7).

## 3.2 MASKED IMAGE MODELING

To alleviate the over-reliance on the noisy pseudo-labels in self-training, we introduce another source of supervision obtained from the raw images. The masked image modeling (MIM) objective aims to learn local image representation at masked positions by predicting the missing information using contextual patches. We follow SimMIM (Xie et al., 2022) and simply predict the RGB pixel values

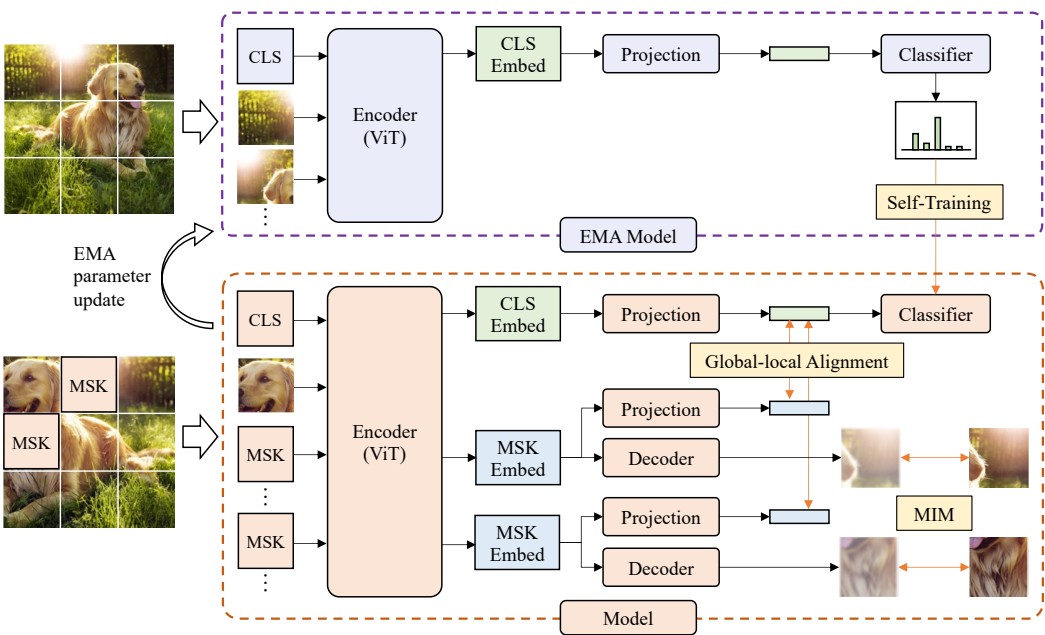

Figure 2: Unsupervised learning framework of MUST. We randomly replace image patches with `[MSK]` tokens, and jointly optimize three objectives on unlabeled images. (1) Global self-training: the classifier uses the output of `[CLS]` to predict the pseudo-label produced by an EMA model. (2) Masked image modeling: the linear decoder uses the outputs of `[MSK]` to predict the pixel values of the masked patches. (3) Global-local feature alignment: minimize the distance between `[CLS]` and `[MSK]` in a normalized embedding space.

for masked patches. Specifically, given the output embedding $z_b^m$ of the $m$-th `[MSK]` token, we first pass it through a linear decoder head to obtain the predicted RGB values $y_b^m \in \mathbb{R}^N$ for that patch, where $N$ denotes the number of RGB pixels per patch. Then we compute the MIM loss as an $\ell_1$-loss between $y_b^m$ and the ground-truth RGB values $x_b^m$:

$$\mathcal{L}_{\text{mim}} = \frac{1}{BMN} \sum_{b=1}^{B} \sum_{m=1}^{M} \|y_b^m - x_b^m\|_1, \tag{4}$$

where $M$ denotes the number of masked patches per image. We employ the patch-aligned random masking strategy (Xie et al., 2022) where multiple $s \times s$ patches are randomly masked with a fix masking ratio for each dataset. In general, we find that a low masking ratio (e.g., 10%) works well for most datasets, which is consistent to the observations in Xie et al. (2022).

## 3.3 GLOBAL-LOCAL FEATURE ALIGNMENT

We aim to bridge the two sources of supervision (*i.e.*, pseudo-labels and image pixels) such that the local `[MSK]` features learned from MIM can improve the global `[CLS]` feature for better classification performance. Let $z_b^c$ denote the output embedding of the `[CLS]` token, and $v_b^c = h(z_b^c)$ denote its normalized embedding after the projection network $h$. Following CLIP, $h$ is a linear layer followed by $\ell_2$ normalization, and $v_b^c$ is subsequently used by the classifier to produce the prediction $p_b$ for self-training (see Section 3.1). We also project the embeddings of the `[MSK]` tokens to the *same* space by passing them through $h$: $v_b^m = h(z_b^m)$. The global-local feature alignment loss is defined as the average squared distance between the normalize embeddings of the `[CLS]` token and all `[MSK]` tokens for each image:

$$\mathcal{L}_{\text{align}} = \frac{1}{BM} \sum_{b=1}^{B} \sum_{m=1}^{M} \|v_b^c - v_b^m\|_2^2 \tag{5}$$

During training, MUST jointly optimizes the above three objectives. The overall loss is

$$\mathcal{L} = \mathcal{L}_{\text{cls}} + \lambda_{\text{reg}} \mathcal{L}_{\text{reg}} + \mathcal{L}_{\text{mim}} + \lambda_{\text{align}} \mathcal{L}_{\text{align}} \tag{6}$$

| Dataset | Train size | Test size | Metric | Description |
|---|---|---|---|---|
| ImageNet (Deng et al., 2009) | 1,281,167 | 50,000 | acc | 1000 categories of objects from WordNet |
| SUN397 (Xiao et al., 2010) | 76,129 | 21,758 | acc | 397 categories of indoor and outdoor scenes |
| Food101 (Bossard et al., 2014) | 75,750 | 25,250 | acc | 101 categories of food dishes |
| GTSRB (Stallkamp et al., 2011) | 26,640 | 12,630 | acc | 43 categories of German traffic signs |
| DTD (Cimpoi et al., 2014) | 3,760 | 1,880 | acc | 47 categories of describable textures |
| UCF101 (Soomro et al., 2012) | 9,537 | 3,783 | acc | 101 categories of human action video frames |
| Oxford Pets (Parkhi et al., 2012) | 3,680 | 3,669 | mean per class | 37 categories of cats and dogs |
| Caltech101 (Binh, 2011) | 3,060 | 6,085 | mean per class | 101 categories of objects and 1 background |

Table 1: Information on the image classification datasets used for transfer learning.

### 3.4 IMPLEMENTATION DETAILS

We experiment with two ViT models pre-trained by Radford et al. (2021): ViT-B/16 and ViT-L/14, containing respectively 12 and 24 Transformer blocks with 768 and 1024 hidden dimensions. The [MSK] token and linear decoder head are randomly initialized and finetuned together with the entire model. During finetuning, we use AdamW (Loshchilov & Hutter, 2017) optimizer with a weight decay of 0.05. We employ a cosine learning rate schedule without any warmup. Following Bao et al. (2022); He et al. (2021), we use a layer-wise learning rate decay (Clark et al., 2020) of 0.65 for both ViT models. The batch size is 1024 for ViT-B/16 and 512 for ViT-L/14, and the learning rate is scaled linearly with the batch size ($lr = $ base_lr $\times$ batchsize$/256$). We use 16 A100 GPUs, and the training process of MUST adds only a small amount of computation overhead after CLIP pre-training. We observe negligible variance on the results between runs with different random seeds.

The model receives input images of size $224 \times 224$. During training, we use RandomResized-Crop+Flip+RandAug (Cubuk et al., 2020) to augment input images, while using Resize+RandomCrop as the weak augmentation to generate pseudo-labels. During test, we simply take a center crop after resizing the shorter edge of the image to 224. For the EMA teacher, we follow Baevski et al. (2022) and linearly ramp-up the parameter decay rate $\mu$ from $\mu_0$ to 0.9998 in $\mu_n$ iterations. Table 10 provides more details of the hyperparameters used for each downstream task, including the pseudo-label threshold, mask patch size, mask ratio, etc.

## 4 EXPERIMENTS

### 4.1 TRANSFER LEARNING DATASETS

We perform experiments on 8 image classification datasets which span many different domains including common objects (Deng et al., 2009; Binh, 2011), fine-grained animals (Parkhi et al., 2012), indoor and outdoor scenes (Xiao et al., 2010), foods (Bossard et al., 2014), traffic signs (Stallkamp et al., 2011), natural textures (Cimpoi et al., 2014), and human actions (Soomro et al., 2012). Table 1 shows the detailed information of each dataset.

### 4.2 MUST RESULTS

Table 2 shows the unsupervised classification results on 6 tasks. MUST substantially improves upon CLIP on all tasks with an average accuracy improvement of **+11.0**% / **+8.5**% for ViT-B / ViT-L. We also experiment with a widely-adopted supervised learning approach: supervised pre-training on ImageNet (1k or 21k) followed by supervised finetuning on the downstream tasks. We finetune the ImageNet model with the same number of epochs and the same data augmentation (Cubuk et al., 2020) as MUST. Compared to the label-intensive supervised learning, unsupervised MUST achieves comparable performance on some datasets (e.g., Food101 and UCF101). MUST still lags behind supervised learning on some datasets (e.g., GTSRB and DTD). We hypothesis that CLIP's low accuracy is the major cause. Since CLIP has not seen enough data from those niche domains during pre-training, its pseudo-labels would contain a limited amount of useful information. Despite this, MUST can narrow the performance gap between unsupervised and supervised learning by around 50%. Therefore, MUST is a low-cost solution for image classification that opens up vast opportunities for practical scenarios where images are abundant but labels are scarce.

| | | ImageNet | SUN397 | Food101 | GTSRB | DTD | UCF101 | Average |
|---|---|---|---|---|---|---|---|---|
| | CLIP | 68.3 | 64.4 | 88.7 | 43.4 | 44.7 | 68.8 | 63.1 |
| ViT-B/16 | MUST | 77.7 (+9.4) | 73.1 (+8.7) | 92.7 (+4.0) | 65.5 (+22.1) | 54.1 (+9.4) | 81.1 (+12.3) | 74.1 (+11.0) |
| | Supervised | 81.8 | 77.2 | 90.3 | 97.1 | 78.6 | 82.3 | 84.5 |
| ViT-L/14 | CLIP | 75.5 | 67.4 | 92.9 | 50.6 | 55.4 | 77.0 | 69.8 |
| | MUST | 82.1 (+6.6) | 75.6 (+8.0) | 95.3 (+2.4) | 68.7 (+18.1) | 62.6 (+7.2) | 85.7 (+8.7) | 78.3 (+8.5) |
| ViT-L/16 | Supervised | 83.4 | 81.1 | 93.1 | 97.3 | 79.7 | 86.4 | 86.8 |

Table 2: Image classification results on a variety of downstream tasks. MUST significantly improves upon CLIP for unsupervised classification on all datasets. Results for CLIP are obtained using the publicly released models. Supervised denotes the label-intensive common practice of supervised pre-training followed by supervised finetuning on downstream tasks. We use the ImageNet-1k model from Touvron et al. (2021) for supervised ViT-B and the ImageNet-21k model from Dosovitskiy et al. (2021) for supervised ViT-L.

| | CLIP | CoCoOp | CoOp | MUST |
|---|---|---|---|---|
| Finetuning data | none | 16 labeled images per class | | unlabeled images |
| ImageNet Acc. | 68.3 | 71.0 | 71.5 | 77.7 |

Table 3: Comparison with few-shot CLIP adaptation methods on ImageNet. All methods use the same CLIP-pretrained ViT-B/16 model as image encoder.

## 4.3 COMPARISON WITH FEW-SHOT ADAPTATION

In Table 3, we compare MUST with two state-of-the-art few-shot adaptation methods (Zhou et al., 2021; 2022) that adapt a pre-trained CLIP model on labeled ImageNet images. MUST outperforms 16-shot adaptation by a large margin (+6.2%), demonstrating its advantage as a low-cost method to improve classification performance using unlabeled data.

## 4.4 UNSUPERVISED TRANSFER ACROSS DOMAINS

Since the [MSK] token and linear decoder are trained from scratch, the potential of MUST may not be fully exploited for downstream tasks with limited number of unlabeled images. To address this, we propose to warm-up the model by training it on a similar domain with more unlabeled images. Specifically, we first em-

| | Pets | Caltech101 |
|---|---|---|
| CLIP | 88.9 | 89.8 |
| MUST w/o ImageNet warmup | 93.1 (+4.2) | 93.1 (+3.3) |
| MUST w/ ImageNet warmup | 94.3 (+5.4) | 93.7 (+3.9) |

Table 4: Warmup on ImageNet improves the performance of MUST on other domains.

ploy MUST to finetune a CLIP ViT-B model on ImageNet for a single epoch, and then continue finetuning the model on two different datasets with limited number of samples (Pets and Caltech101). During warmup, we freeze the linear classifier (i.e., CLIP's text embeddings) to anchor the normalized image embeddings in their original space for easier transfer. As shown in Table 4, warmup on ImageNet further improves the performance of MUST on both datasets.

## 4.5 ABLATION STUDY

**Effect of the proposed objectives.** We study the effect of the three objectives: self-training, mask image modeling, and global-local feature alignment. The results are shown in Table 5. Removing the alignment loss reduces the average accuracy by 1.5%, and removing MIM further decreases accuracy by 0.7%. The results demonstrate the importance to align the features of [CLS] and [MSK] tokens, which regularizes the model to learn better global features for classification.

**Number of [MSK] tokens to align with.** Our global-local feature alignment loss aims to align the [CLS] token to all of the [MSK] tokens for an image (see equation 5). We relax the alignment strength by only using the 10 [MSK] tokens that are nearest to the [CLS] token in the embedding space. As shown in Table 6, using the nearest-10 [MSK] tokens leads to similar performance compared to using all [MSK] tokens, which suggests that MUST is robust to the strength of alignment.

| MUST objectives | ImageNet | SUN397 | Food101 | GTSRB | DTD | UCF101 | Average |
|---|---|---|---|---|---|---|---|
| ST+MIM+Align | 77.7 | 73.1 | 92.7 | 65.5 | 54.1 | 81.1 | 74.1 |
| ST+MIM | 77.1 | 72.0 | 92.4 | 62.5 | 52.6 | 79.2 | 72.6 (–1.5) |
| ST | 76.5 | 71.4 | 92.4 | 59.8 | 52.4 | 79.0 | 71.9 (–2.2) |

Table 5: Effect of the proposed training objectives on unsupervised image classification.

| Num. [MSK] for $\mathcal{L}_{align}$ | ImageNet | SUN397 | Food101 | GTSRB | DTD | UCF101 | Average |
|---|---|---|---|---|---|---|---|
| All | 77.7 | 73.1 | 92.7 | 65.5 | 54.1 | 81.1 | 74.1 |
| Nearest-10 | 77.6 | 73.2 | 92.7 | 65.7 | 54.1 | 80.8 | 74.1 |

Table 6: Number of [MSK] tokens used to align with the [CLS] token.

| | ImageNet | SUN397 | Food101 | GTSRB | DTD | UCF101 | Average |
|---|---|---|---|---|---|---|---|
| ST w/ $\mathcal{L}_{reg}$ | 76.5 | 71.4 | 92.4 | 59.8 | 52.4 | 79.0 | 71.9 |
| ST w/o $\mathcal{L}_{reg}$ | 74.0 | 71.0 | 91.8 | 50.0 | 46.8 | 73.0 | 67.8 (–4.1) |

Table 7: Effect of the fairness regularization loss on self-training performance.

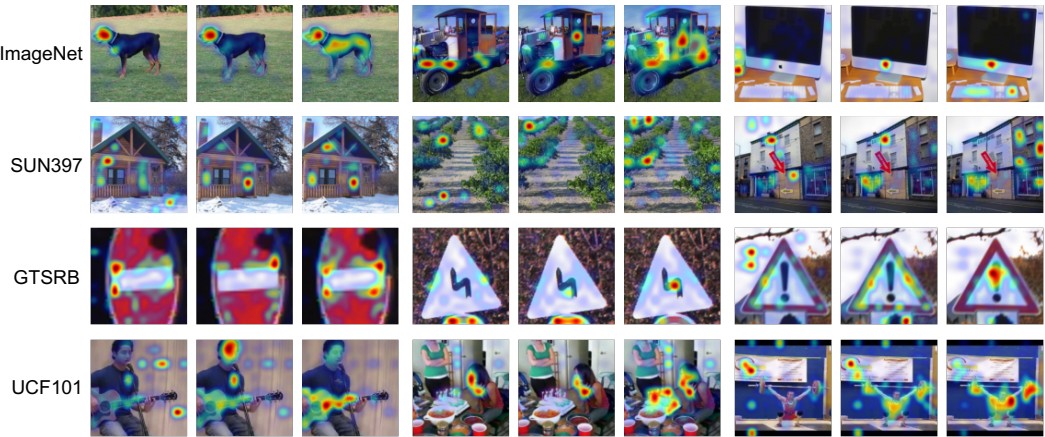

Figure 3: GradCAM visualization on validation images from multiple datasets. We visualize the self-attention from [CLS] to all patches in the last ViT layer. For each triplet, we compare three different models: CLIP (left), ST (middle), MUST (right). We make two qualitative observations: (1) MUST generally relies on more patches to make predictions; (2) MUST can better leverage the true correlation between patches and class labels.

**Effect of fairness regularization.** We propose the fairness regularization loss $\mathcal{L}_{reg}$ to counteract the bias in pseudo-labels. In Table 7, we examine its effect on the self-training objective. The regularization improves the performance of self-training on all datasets. It is the least helpful on SUN397, which has a long-tailed class distribution in its training set (Xiao et al., 2010).

## 4.6 QUALITATIVE ANALYSIS

**MUST pays attention to more informative regions.** In Figure 3, we visualize the GradCAM (Selvaraju et al., 2017) heatmap for the self-attention from [CLS] to all patches in the last ViT layer. We compare three models - CLIP, ST, MUST- and make the following qualitative observations: (1) MUST generally relies on more patches to make predictions. E.g., the fully body of the dog instead of only the head in row 1 column 1; all four corners of the traffic sign in row 3 column 1. (2) MUST can better leverage the true correlation between image patches and class labels, instead of the spurious correlations used by CLIP and ST. E.g., the guitar is attended for the action "playing guitar" in row 4 column 1, instead of the person or the background.

**A helpful MIM is not necessarily good at image recovery.** In Figure 4, we show examples of recovered images from the MIM output on the validation set of different datasets. On some datasets

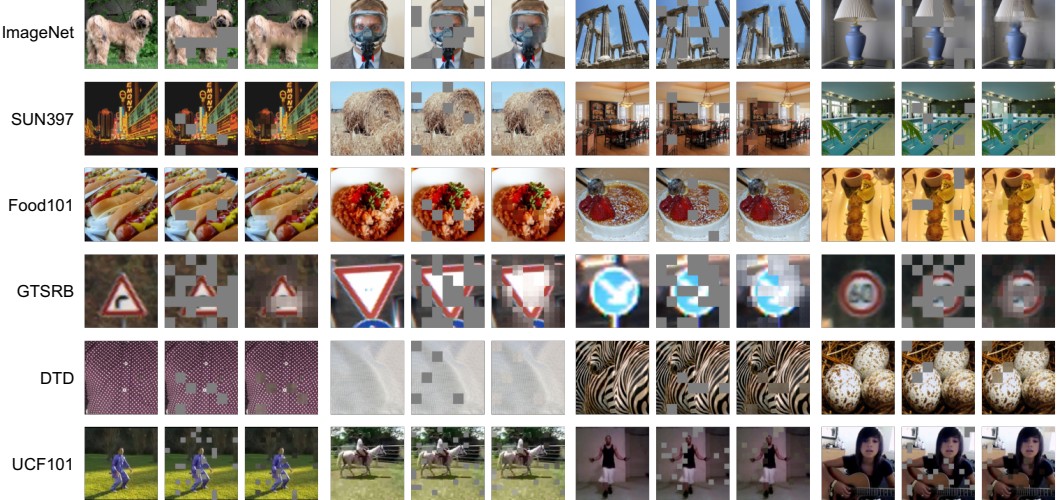

Figure 4: Examples of mask image modeling on validation images from multiple datasets. For each triplet, we show the original image (left), the masked image (middle), and the recovered image (right).

| ImageNet-C Hendrycks & Dietterich (2019) | CLIP | ST | MUST | MAE+Supervised He et al. (2021) |
|---|---|---|---|---|
| mCE ↓ | 70.7 | 55.2 | 53.4 | 51.7 |

Table 8: Robustness evaluation on the ImageNet-Corruption dataset.

| | ImageNetV2 | ImageNet-R | ImageNet-A |
|---|---|---|---|
| CLIP | 61.9 | 77.7 | 49.9 |
| Inductive MUST on ImageNet | 68.9 (+7.0) | 68.7 (−9.0) | 41.6 (−8.3) |
| Transductive MUST w/o ImageNet warmup | 64.7 (+2.8) | 87.3 (+9.6) | 56.9 (+7.0) |
| Transductive MUST w/ ImageNet warmup | 66.9 (+5.0) | 87.6 (+9.9) | 58.2 (+8.3) |

Table 9: Evaluation of MUST under natural distribution shift from ImageNet.

with abundant samples (e.g., ImageNet), the model can learn to recover masked patches with high quality. On other datasets with fewer number of samples (e.g., GTSRB), the recovered images have lower quality with many artifacts. Despite the low recovery quality, the MIM objective still provides useful supervision that improves the image classification performance.

## 4.7 ROBUSTNESS TO IMAGE CORRUPTIONS

In Table 8, we study the robustness of MUST to image corruptions. We evaluate the performance of ImageNet-finetuned models on the ImageNet-C(Hendrycks & Dietterich, 2019) benchmark, which contains 15 corruption types with various severity. MUST is more robust than CLIP and ST, as suggested by a lower mean corruption error (mCE). The mCE of MUST is only slightly higher than a model that is first trained with self-supervised MAE (He et al., 2021) followed by *supervised* finetuning on ImageNet.

## 4.8 TRANSDUCTIVE ADAPTATION TO DISTRIBUTION SHIFT

We further investigate the robustness of MUST under natural distribution shifts. We experiment with three out-of-distribution variants of ImageNet: ImageNetV2 (Recht et al., 2019) with the same 1000 ImageNet classes, ImageNet-Rendition (Hendrycks et al., 2021a) and ImageNet-Adversarial (Hendrycks et al., 2021b) where each contains a subset of 200 classes (see Figure 5 for example images). First, we directly evaluate a ViT-B model finetuned on ImageNet using MUST, which has 77.7% accuracy on ImageNet validation set. As shown in Table 9, MUST on ImageNet improves the classification accuracy on ImageNetV2, but decreases the performance on ImageNet-R and ImageNet-A. A similar observation is reported in CLIP (Radford et al., 2021), where supervised adaptation to ImageNet reduces the model's average robustness to out-of-distribution datasets.

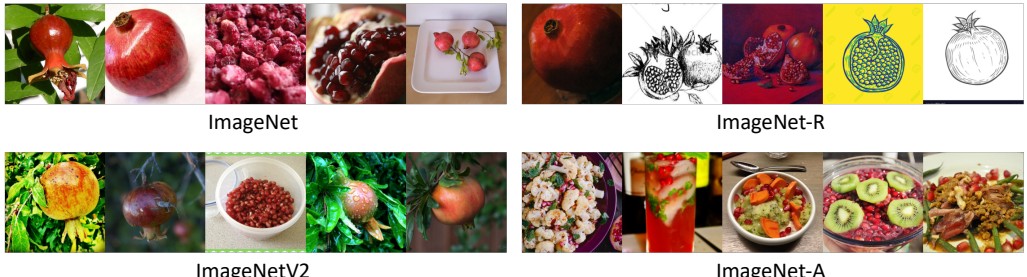

Figure 5: Example images of *pomegranate* from ImageNet and other datasets of natural distribution shifts.

| | ImageNet | | SUN397 | | Food101 | | GTSRB | | DTD | | UCF101 | | Pets | Caltech101 |
| | B | L | B | L | B | L | B | L | B | L | B | L | B | B |
|---|---|---|---|---|---|---|---|---|---|---|---|---|---|---|
| base lr | 2e-5 | 4e-5 | 1e-4 | 2e-4 | 2e-5 | 4e-5 | 2e-5 | 4e-5 | 2e-5 | 4e-5 | 1e-5 | 2e-5 | 1e-5 | 1e-5 |
| epoch | 30 | 20 | 20 | 20 | 40 | 40 | 20 | 20 | 60 | 60 | 60 | 60 | 60 | 60 |
| threshold $\tau$ | 0.7 | 0.5 | 0.4 | 0.3 | 0.7 | 0.7 | 0.3 | 0.2 | 0.4 | 0.5 | 0.5 | 0.6 | 0.7 | 0.7 |
| mask ratio | 0.3 | 0.3 | 0.1 | 0.1 | 0.1 | 0.1 | 0.5 | 0.3 | 0.1 | 0.1 | 0.1 | 0.1 | 0.1 | 0.1 |
| mask patch size | 32 | 56 | 32 | 28 | 32 | 56 | 32 | 56 | 32 | 56 | 16 | 14 | 32 | 32 |
| $\lambda_{\text{align}}$ | 0.2 | 0.5 | 1 | 0.5 | 0.1 | 0.5 | 0.5 | 0.5 | 0.2 | 0.5 | 0.5 | 0.5 | 0.1 | 0.1 |
| $\lambda_{\text{reg}}$ | 1 | | 0.2 | | 1 | | 1 | | 1 | | 1 | | 1 | 1 |
| EMA init. $\mu_0$ | 0.999 | | 0.99 | | 0.999 | | 0.99 | | 0.99 | | 0.99 | | 0.99 | 0.99 |
| EMA iters. $\mu_n$ | 2000 | | 500 | | 2000 | | 500 | | 100 | | 500 | | 100 | 100 |

Table 10: Hyperparameters for MUST on the downstream datasets. B: ViT-B/16; L: ViT-L/14.

To address this, we perform transductive transfer learning (Xian et al., 2017; Rohrbach et al., 2013) with MUST. Transductive learning assume access to unlabeled test images from the new distribution, which MUST uses to jointly update model parameters and infer image labels. The results in Table 9 show that transductive MUST substantially improves the model's accuracy to distribution shift. Furthermore, the warmup on ImageNet (Section 4.4) before transductive learning leads to more improvement.

## 5  LIMITATIONS

In this paper, we show MUST as a promising domain-specific label-free classification algorithm where a different model is trained for each task. Although the adaptation process is efficient, the storage of additional model parameters could be a practical limitation if the number of tasks is large. There exists a simple way to address this concern: gather unlabeled image from all the domains of interest, and perform MUST to learn a single model that can generalize to multiple domains. However, this may sacrifice the model's performance on each individual domain.

Due to the use of CLIP pre-trained models, MUST inherits some of the social implications of CLIP that arose from using minimally-filtered data crawled on the open internet. Additional analysis on the model is necessary before deploying it for real-world applications.

## 6  CONCLUSION

The field of deep learning has been largely driven by simple and effective methods that can scale well. MUST is a simple method that can effectively adapt a pre-trained zero-shot classifier to downstream tasks in an unsupervised manner. The core of MUST is to simultaneously leverage two different sources of supervision signals obtained from unlabeled data: task-specific supervision from pseudo-labels and task-agnostic self supervision from raw images. We propose a simple alignment objective that bridges the two sources of supervision and enables the model to learn better features.

As a future work, MUST can be naturally extended to other domains such as NLP, where mask language modeling could act as the task-agnostic objective to improve self-training. We hope that this perspective could inspire future methods in unsupervised learning and zero-shot classification.

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
