# OpenReview forum: "Masked Unsupervised Self-training for Label-free Image Classification "
_ICLR.cc/2023/Conference — ICLR 2023 poster_

### Official Review · Reviewer_rbuG · 2022-10-13

**Confidence:** 3
**Correctness:** 3
**Technical Novelty And Significance:** 3
**Empirical Novelty And Significance:** 3
**Recommendation:** 8

**Clarity, Quality, Novelty And Reproducibility:**

Overall the paper is very clear, novel, and various hyperparameter details are provided which makes it seem reproducible.

**Strength And Weaknesses:**

Strengths:
- This paper appears to be a very detailed, thorough, and reproducible empirical investigation and the results are potentially very useful to the community.
- Extensive ablations are performed on the design decisions.

Weaknesses:
- I found the main weakness to be the experimental set-up - I believe this would be much more convincing if the paper considered a datasets which was not clean/balances such as LAION or CC12m for running their procedure.
- Some comparisons seem to be a bit unfair, for instance Table 3 in which MUST (seeing all images) is compared to methods seeing 16 images per class, what if MUST only sees 16 images per class.

Questions:
- What happens if you replace the self-training part with the real labels? Are the other losses still required?

**Summary Of The Paper:**

This paper proposes a method of adapting a zero-shot model to a dataset such as ImageNet without using any labels from ImageNet. This process does improve model accuracy on ImageNet by a large margin, which is perhaps not too surprising as the target dataset is used for this adaptation. Their adaptation method consists mainly of masked image modeling and self-training from an EMA teacher.

**Summary Of The Review:**

Overall I enjoyed this paper, which is thorough and observes good empirical performance. My main concern is in the experimental set-up which takes a clean, balanced data-set then removes the labels.

---

> ### Author Response · Authors · 2022-11-15
> **Authors' Response**
>
> We appreciate the reviewer's positive comments. Next we address the reviewer’s questions.
>
> > I believe this would be much more convincing if the paper considered a datasets which was not clean/balances such as LAION or CC12m for running their procedure
>
> Since MUST performs self-training using pseudo-labels as supervision, it is required that the unlabeled images share the same set of classes as the target task. Further extensions to use generic image-text pair datasets (e.g., LAION) is possible though likely non-trivial, which we can explore as future work.
>
> We would also like to clarify that MUST does not require balanced datasets. One of our training dataset, SUN397, has a long-tailed distribution. MUST achieves substantial performance improvement nonetheless.
>
>
> > Some comparisons seem to be a bit unfair, for instance Table 3 in which MUST (seeing all images) is compared to methods seeing 16 images per class, what if MUST only sees 16 images per class.
>
> The few-shot methods in Table 3 sees *labeled* images, whereas MUST only sees *unlabeled* images. Therefore, we argue that MUST is at a disadvantage compared to few-shot methods. We leave it as future work to extend MUST for few-shot or semi-supervised settings.
>
>
> > What happens if you replace the self-training part with the real labels? Are the other losses still required?
>
> Given a large number of labeled images, one can simply perform supervised learning without additional losses. However, supervised learning incurs expensive annotation costs, which motivates us to develop unsupervised learning methods. MUST effectively addresses the unique challenges in unsupervised learning for label-free image classification.

---

> > ### Comment · Reviewer_rbuG · 2022-11-15
> > **Thanks**
> >
> > Thanks for the response.

---

### Official Review · Reviewer_Xc1a · 2022-10-15

**Confidence:** 4
**Correctness:** 3
**Technical Novelty And Significance:** 3
**Empirical Novelty And Significance:** 3
**Recommendation:** 8

**Clarity, Quality, Novelty And Reproducibility:**

Clarity: Good clarity, nice presentation.

Quality: Good paper, which clearly meets the acceptance bar of ICLR.

Novelty: Not very novel. Incremental on MIM and self-training.

Reproducibility: Training details and codes are provided.

**Details Of Ethics Concerns:**

Ethics issues are discussed.

**Strength And Weaknesses:**

Strength:

1: Clear motivation. This paper is well-motivated.

2: Good writing and presentation, easy to follow/read.

3: Very solid experiments and ablation study. Multiple datasets are evaluated. Robustness and calibration metrics are reported.

4: Good performance. The proposed method surpasses CLIP by large margins.

Main weaknesses:

1: The best feature of CLIP is the generality, that is, being able to recognize any image without pre-defined/fixed classes. MUST adapt the CLIP model to a specific dataset (which is the main purpose of this paper). A simple solution is claimed by authors in the limitation section: "There exists a simple way to address this concern: gather unlabeled image from all the domains of interest, and perform MUST to learn a single model that can generalize to multiple domains."  It would be really great to have 1-2 such experiments to verify this hypothesis.

Minor weaknesses:

2:  The paper does not reach out to theoretical backup to explain why MUST works.

3: Adding results on ImageNet-Sketch will further strengthen this paper.

4: A related work [1] is worth discussing.

[1]: Test-time training with masked autoencoders, NeurIPS 2022


**Summary Of The Paper:**

This paper introduces MUST (MASKED UNSUPERVISED SELF-TRAINING) to improve the performance of CLIP-like models in zero-shot recognition. Based on a pre-trained CLIP, three objectives are learned together to fine-tune it on a specific dataset in an unsupervised manner. Three proposed objectives are class-level global feature and pixel-level local feature and enforce a regularization between the previous two objectives. A fairness regularization loss is introduced in the class-level global feature objective.
Experiments are conducted on 8 common datasets and the results clearly demonstrate the superiority of the proposed methods, which surpass baseline CLIP by large margins. The ablation study is solid, and the individual contribution of each component is demonstrated.

**Summary Of The Review:**

The authors addressed all of my concerns during the discussion. I'm leaning toward acceptance.

---

> ### Author Response · Authors · 2022-11-15
> **Authors' Response**
>
> We appreciate the reviewer's positive comments. Next we address the reviewer’s questions.
>
> > Training a unified MUST model on multiple domains
>
> Following the reviewer’s suggestion, we extend MUST to the multi-domain setting by training a unified model on unlabeled images from multiple datasets.  Different domains share the same backbone ViT, and each domain has its individual classifier-head for domain-specific classification. We choose three downstream datasets for this experiment: SUN397, Food101, and UCF101. The results are shown in the following Table. It verifies our hypothesis: a unified MUST model is more parameter-efficient, but leads to smaller improvement over CLIP compared to domain-specific MUST models.
>
> Method  | SUN397 | Food101 | UCF101 |
> | --- | --- | --- | --- |
> CLIP | 64.4 | 88.7 | 68.8 |
> MUST | 73.1 | 92.7 | 81.1 |
> MUST w/ shared backbone | 69.5 | 91.0 | 75.9 |
>
> > Results on ImageNet-Sketch
>
> Our paper has experimented with three out-of-distribution variants of ImageNet, which we believe are sufficient to demonstrate the robustness of MUST. Nevertheless, we have carried out additional experiments on ImageNet-sketch as the reviewer suggested. The results are shown in the following table. The improvement from transductive MUST is consistent with the findings in our paper.
>
> Method  | Accuracy
>  --- | ---
> CLIP | 48.2
> Inductive MUST on ImageNet | 44.5 (-3.7)
> Transductive MUST w/o ImageNet warmup | 56.3 (+8.1)
> Transductive MUST w/ ImageNet warmup | 57.4 (+9.2)
>
>
> > The paper does not reach out to theoretical backup to explain why MUST works.
>
> Our paper focuses on the practical contribution of MUST which enables zero-shot classifiers to meet the performance criteria for real-world deployment. The design of our framework is motivated by theoretical principles from self-training and self-supervised learning, such as the multi-view principle [a]. We leave it as future work for rigorous theoretical study of MUST.
>
> [a] Towards Understanding Ensemble, Knowledge Distillation and Self-Distillation in Deep Learning. Allen-Zhu et al., 2022.
>
>
> > A related work [1] is worth discussing
>
> We appreciate the reviewer’s suggestion and have updated the related work section of our paper.

---

> > ### Comment · Reviewer_Xc1a · 2022-11-15
> > **Response to authors1603**
> >
> > Thanks for preparing your rebuttal (which was very carefully crafted -- good job on that)! It's great to see experimental results regarding MUST on multiple domains. This rebuttal does fully address my initial concerns about this paper.
> > It is a good paper that may have significant contributions to the field, I will raise my rating to accept.

---

### Official Review · Reviewer_zxPa · 2022-10-25

**Confidence:** 3
**Correctness:** 3
**Technical Novelty And Significance:** 3
**Empirical Novelty And Significance:** 3
**Recommendation:** 8

**Clarity, Quality, Novelty And Reproducibility:**

The paper written in a clear and direct fashion.

One small nit: it's not clear that Fig. 2 needs 2-3 different font styles.



**Strength And Weaknesses:**

The paper's proposal is mostly clear well-motivated.

Experiments are well thought out and clear and show a clear advantage over direct usage of CLIP across a variety of downstream
tasks. Many ablations probe the inner working of the loss and analyze what each factor is doing.

A weakness could be the global-local loss, which I felt was the least grounded in any kind of motivation as it is the "average squared distance between the normalize embeddings of the [CLS] token and all [MSK] tokens for each image". However, this seems like this decision could be a bias towards datasets that contain a single large and well-framed object in the field of view.




**Summary Of The Paper:**

The paper proposes Masked Unsupervised Self-Training (MUST), that uses pseudo labels from CLIP as well as patch masking to train on unlabeled data. The loss consists of three terms a) self training classification loss, b) a masked patch loss, and c) a global-local loss designed to force class level information into the patch embeddings.




**Summary Of The Review:**

The paper clearly proposes a method of performing self supervised training using CLIP and a pool of unlabeled data by constructing a loss from classification losses and masked patch losses. A wide variety of experiments are performed over a set of datasets that show the contributions of the various components of MUST.

---

> ### Author Response · Authors · 2022-11-15
> **Authors' Response**
>
> We appreciate the reviewer's positive comments. As the reviewer suggested, our method achieves a clear advantage over CLIP with a simple framework, where the effect of each proposed component is thoroughly probed. Next we address the reviewer's questions.
>
> > Motivation of the global-local loss.
>
> The proposed global-local alignment loss aims to bridge the knowledge learned from the global pseudo-label and the local image pixels. It forces the [CLS] token to contain local image information, thus mitigating overfitting to pseudo-labels. Its effect can be seen in Figure 3, where the [CLS] token relies on more image patches to make predictions.
>
> > The global-local loss could be a bias towards datasets that contain a single large and well-framed object in the field of view.
>
> From our ablation study in Table 5, the global-local alignment loss brings consistent improvement on various datasets, including SUN397 (scene classification) and USF101 (video action classification), where the images often do contain a single large object.
>
> > It's not clear that Fig. 2 needs 2-3 different font styles.
>
> We appreciate the valuable suggestion and have revised the figure accordingly.

---

### Official Review · Reviewer_SBFM · 2022-10-25

**Confidence:** 4
**Correctness:** 4
**Technical Novelty And Significance:** 2
**Empirical Novelty And Significance:** 3
**Recommendation:** 8

**Clarity, Quality, Novelty And Reproducibility:**

Clarity: Clear and easy to understand paper
Quality: Significant improvements over the baseline of CLIP and a thorough ablation study
Novelty: Limited, with the proposed loss being a sum of two existing methods (self training and masked image modeling) and a novel alignment term.
Reproducibility: The proposed methods is fairly straightforward, with code in supplementary.



**Strength And Weaknesses:**

Strengths
- Clear and easy to understand writing
- Simple method.
- Significant performance improvements over CLIP based baseline approach.
- Meaningful and informative ablation experiments that study various aspects of the proposed approach.

Weaknesses
- Limited technical novelty, with the first two terms of the proposed loss already existing and the alignment term being the primary technical contribution.

**Summary Of The Paper:**

Paper proposes a method, MUST, for unsupervised adaptation of a zero shot classifier. A pre-trained open world model CLIP is taken to produce classification embeddings for a vocabulary of words. These are then use to kick-start a training method that adapts a zero-shot classification model by a combination of three objectives. The first one uses a self-training, second uses a masked image modeling objective and the third one enforces global local feature alignment.The proposed method is then evaluated on a variety of datasets.

**Summary Of The Review:**

The paper proposes a simple methods (though limited in technical novelty), that leads to significant improvements in zero shot results. I vote accept.

---

> ### Author Response · Authors · 2022-11-15
> **Authors' Response**
>
> We appreciate the reviewer's positive comments. As the reviewer suggested, our method achieves significant performance improvement with a simple framework, where the effect of each proposed component is thoroughly investigated.
>
> Our novelty resides in the overall label-free classification framework which exploits two complementary sources of supervision from unlabeled images. To fulfil our proposed framework, we leverage existing training objectives when possible and propose the new alignment objective to bridge the gap between the two supervision signals.

---

### Official Review · Reviewer_RF4U · 2022-10-28

**Confidence:** 4
**Clarity, Quality, Novelty And Reproducibility:** There is nothing obviously wrong with…
**Correctness:** 4
**Technical Novelty And Significance:** 3
**Empirical Novelty And Significance:** 3
**Recommendation:** 5

**Strength And Weaknesses:**

1. Strength:
The authors propose a very effective method for label-free image classification. The performance is impressive in different benchmarks.

2. weakness:
However, I think novelty is debatable. Frankly, it looks like three different self-supervised methods are being used. EMA + L2 loss is BYOL [1], local-global contrast is deep informax [3] and masked autoencoder [2]. And obviously it losts the [3] in the reference.


[1]. Bootstrap your own latent: A new approach to self-supervised Learning
[2]. Masked Autoencoders Are Scalable Vision Learners.
[3]. Learning deep representations by mutual information estimation and maximization.

**Summary Of The Paper:**

In this paper, the authors combine three different self-supervised methods into label-free image classification: Infomax, simsiam(BYOL) and masked encoder. In experiments, the results demonstrate significant improvement in different benchmarks.

**Summary Of The Review:**

Because of novelty, I tend to give rejection to this paper. However I am happy to refer to other reviewers as I am only an expert in self-supervised learning and not CLIP related methods.

---

> ### Author Response · Authors · 2022-11-15
> **Authors' Response**
>
> We appreciate the reviewer's constructive comments. Here we address the review’s questions.
>
> First of all, our proposed MUST tackles a fundamentally different problem compared to self-supervised representation learning [1,2,3]. MUST trains a *task-specific* classification model using unlabeled images, whereas self-supervised learning trains a *task-agonistic* model that requires an additional stage of fine-tuning for classification tasks. Therefore, MUST addresses unique challenges for label-free image classification that do not exist in previous self-supervised learning papers.
>
> MUST unlocks the potential of vision-language pre-trained models to be practically useful on downstream classification tasks. The significance of our contribution is confirmed by the positive comments from other reviewers:
> - Reviewer 8jDS: "The paper provides a significant contribution to exploring methods of overcoming the current issues we have with the cost of labelling datasets."
> - Reviewer SBFM: "Significant performance improvements over CLIP based baseline approach."  Reviewer zxPa: "... show a clear advantage over direct usage of CLIP across a variety of downstream tasks." Reviewer Xc1a: "The proposed method surpasses CLIP by large margins."
> - Reviewer rbuG: "The results are potentially very useful to the community."
>
> > EMA + L2 loss is BYOL.
>
> Our self-training uses a task-specific cross-entropy classification loss, which is different from the task-agnostic L2 loss from BYOL.
>
> > local-global contrast is deep infomax [3]
>
> We appreciate the reviewer for the Deep InfoMax (DIM) paper [3] and have added its reference in our paper. However, our proposed global-local alignment loss is different from the DIM loss [3] both *conceptually* and *implementation-wise*.
>
> Conceptually, DIM aims to achieve task-agnostic self-supervised representation learning. Different from DIM, our global-local alignment loss aims to improve the classification performance. It forces the [CLS] embedding to contain local image information, thus mitigating overfitting to pseudo-labels.
>
> Implementation-wise, DIM requires negative samples and an additional MI estimator. On the contrary, our global-local alignment loss is a simple MSE loss without the need of negative samples. In fact, we have experimented with the contrastive loss (InfoNCE) and observed worse performance compared to the simpler MSE loss.
>
> We hope that our response can clear up the reviewer’s concerns and consider raising the score.

---

### Official Review · Reviewer_8jDS · 2022-10-28

**Confidence:** 4
**Correctness:** 4
**Technical Novelty And Significance:** 3
**Empirical Novelty And Significance:** 3
**Recommendation:** 8

**Clarity, Quality, Novelty And Reproducibility:**

Paper is clear and well written. Concepts are introduced and explained really well to allow a reader to be able to seamlessly follow along as they read. The experimental procedures was well articulated and sounds reproducible.

**Strength And Weaknesses:**

Strengths
- The paper is well written and concepts are explained well.
- Given the high costs of labelling datasets, the authors explore alternative methodologies that try to overcome this barrier.

Weaknesses
- Given the authors' awareness of the negative social implications of CLIP pretrained models, what steps will/did you take to minimize these effects given that you will possibly share your trained models?

**Summary Of The Paper:**

The paper introduces a new method of using machine learning to classify images without the use of labels. Training consists of 3 parts:
- The self training part where the teacher model generates pseudo-labels to train the student model to learn global tasks.
- The masked image modelling part where they train a model to predict missing pixels of an image in order to learn how to represent local pixels information.
- The part where they connect the knowledge learned from the two models in order to do the classification.
They use this method to investigate how well it performs on 8 image classification tasks and compares its efficacy to other models like CLIP that are known to perform really well in this field. MUST outperforms CLIP based models and seems to be using the relevant/defining parts of the image that correlate with the label to make the classification.

**Summary Of The Review:**

Overall, I think the paper provides a significant contribution to exploring methods of overcoming the current issues we have with the cost of labelling datasets. The methodology is promising and I would look forward to future work extending it to apply to more diverse real world datasets.

---

> ### Author Response · Authors · 2022-11-15
> **Authors' Response**
>
> We appreciate the reviewer's positive comments. As the reviewer suggested, our paper provides a significant contribution to overcome the cost of labelling datasets.
>
> > Given the authors' awareness of the negative social implications of CLIP pretrained models, what steps will/did you take to minimize these effects given that you will possibly share your trained models?
>
> We thank the reviewer for raising this important question. Our proposed method contains a fairness regularization loss that can mitigate the bias from CLIP pre-trained models. For deploying our method in real-world applications, we also encourage a manual inspection of the images used for fine-tuning.

---

> > ### Comment · Reviewer_8jDS · 2022-11-17
> > **Thank you**
> >
> > Thank you for the response.

---

### Decision · Program_Chairs · 2023-01-20

**Decision:**

Accept: poster

**Justification For Why Not Higher Score:**

Most of reviewers liked the paper but also judging from their reviews, they did not challenge the paper in terms of its theoretical foundations. The lack of deep theoretical analysis for the choices and somehow incremental design as noted by one of reviewers (EMA + L2 loss is BYOL [1], local-global contrast is deep informax [3] and masked autoencoder [2]) make it hard for AC to recommend a spotlight. Masked learning, EMA etc. are all standard techniques frequently used currently in many papers. While the use of the above tools is interesting, the paper still lacks the necessary analytical depth to be recommended for a spotlight.

**Justification For Why Not Lower Score:**

Overall reviewers were happy with the paper and did not voice any major issues.

**Metareview: Summary, Strengths And Weaknesses:**

Overall, all reviewers have been in favor of this submission but also some number of concerns showed up. For example, lack of theoretical analysis for the choices and somehow incremental design (EMA + L2 loss is BYOL [1], local-global contrast is deep informax [3] and masked autoencoder [2]). While the use of the above tools is interesting, the paper still lacks the necessary analytical depth to be recommended for a spotlight.

**Note From Pc:**

if the above contains the word "oral" or "spotlight" please see: "oral" presentation means -> notable-top-5% and "spotlight" means -> notable-top-25%. As stated in our emails, we are disassociating presentation type from AC recommendations

**Summary Of Ac-Reviewer Meeting:**

Not a borderline paper.